# Methylation of *SRD5A2* promoter predicts a better outcome for castration-resistant prostate cancer patients undergoing androgen deprivation therapy

**Zongwei Wang**[1,2☯], **Tuo Deng**[1,3☯], **Xingbo Long**[4], **Xueming Lin**[2,5], **Shulin Wu**[2], **Hongbo Wang**[2], **Rongbin Ge**[6], **Zhenwei Zhang**[7], **Chin-Lee Wu**[2], **Mary-Ellen Taplin**[7], **Aria F. Olumi**[1]*

**1** Department of Surgery, Division of Urology, Beth Israel Deaconess Medical Center, Harvard Medical School, Boston, MA, United States of America, **2** Department of Pathology and Urology, Massachusetts General Hospital, Harvard Medical School, Boston, MA, United States of America, **3** Department of Urology, Minimally Invasive Surgery center, The First Affiliated Hospital of Guangzhou Medical University, Guangdong Key Laboratory of Urology, Guangzhou Institute of Urology, Guangzhou, Guangdong, China, **4** Department of Urology, Union Medical College, Beijing, China, **5** Department of Urology, First Hospital of Shanxi Medical University, Taiyuan, Shanxi, China, **6** Department of Pathology, University of Massachusetts Medical School, Worcester, MA, United States of America, **7** Dana-Farber Cancer Institute, Harvard Medical School, Boston, MA, United States of America

☯ These authors contributed equally to this work.
* aolumi@bidmc.harvard.edu

**Data Availability Statement:** All relevant data are within the paper and its Supporting Information files.

## Abstract

### Purpose

To determine whether *SRD5A2* promoter methylation is associated with cancer progression during androgen deprivation therapy (ADT) in CRPC.

### Patients and methods

In a Local CRPC cohort, 42 prostatic specimens were collected from patients who were diagnosed as CRPC and underwent transurethral resection of the prostate (TURP) at Massachusetts General Hospital (MGH). In a metastatic CRPC (Met CRPC) cohort, 12 metastatic biopsies were collected from CRPC patients who would be treated with abiraterone plus dutasteride (Clinical Trial NCT01393730). As controls, 36 benign prostatic specimens were collected from patients undergoing prostate reduction surgery for symptoms of bladder outlet obstruction secondary to benign prostatic hyperplasia (BPH). The methylation status of cytosine-phosphate-guanine (CpG) site(s) at *SRD5A2* promoter regions was tested.

### Results

Compared with benign prostatic tissue, CRPC samples demonstrated higher *SRD5A2* methylation in the whole promoter region (Local CRPC cohort: $P < 0.001$; Met CRPC cohort: $P < 0.05$). In Local CRPC cohort, a higher ratio of methylation was correlated with better OS ($R2 = 0.33$, $P = 0.013$). Hypermethylation of specific regions (nucleotides -434 to -4 [CpG# -39 to CpG# -2]) was associated with a better OS (11.3±5.8 vs 6.4±4.4 years, $P = 0.001$) and PFS (8.4±5.4 vs 4.5±3.9 years, $P = 0.003$) with cutoff value of 37.9%. Multivariate

**Funding:** AFO gratefully acknowledges financial support from NIH/NIDDK (NIH/R01 DK091353). ZW was supported by the Urology Care Foundation/American Urological Association Research Scholar Award.

**Competing interests:** The authors have declared that no competing interests exist.

analysis showed that *SRD5A2* methylation was associated with OS independently (whole promoter region: P = 0.035; specific region: *P* = 0.02).

## Conclusion

Our study demonstrate that *SRD5A2* methylation in promoter regions, specifically at CpG# -39 to -2, is significantly associated with better survival for CRPC patients treated with ADT. Recognition of epigenetic modifications of *SRD5A2* may affect the choices and sequence of available therapies for management of CRPC.

## Introduction

Advanced castration resistant prostate cancer (CRPC) accounts for majority of 31,000 deaths each year in the United States, and prognostic tools that determine patients' overall survival (OS) are lacking [1, 2]. Oral inhibitors targeting CYP-17 (by abiraterone) and the androgen receptor (AR) (by enzalutamide, apalutamide, darolutamide) have increased survival in CRPC in phase III studies [3–8]. However, resistance to AR-directed therapies remains a challenge, which indicates a complexity in the progression from invasive cancer to castration-resistant disease. Persistent AR signaling despite AR-axis inhibition is a critical mechanism of resistance in patients with metastatic CRPC (Met CRPC) [9]. Thus, a better understanding of the drivers in resistance is needed to develop therapeutic strategies that offer patients long-term clinical benefit.

Predictive biomarkers identifying CRPC patients who respond optimally to androgen deprivation therapy (ADT) is of great clinical importance. One mechanism of CRPC development is intratumoral biosynthesis of dihydrotestosterone (DHT) from adrenal precursors, which requires three enzymatic steps. 5-α reduction catalyzed by steroid 5-α reductase (SRD5A) is one of the three steps [10]. SRD5A has three isoforms, SRD5A1, SRD5A2 and SRD5A3. A recent study demonstrated that the expression of SRD5A1 and SRD5A2 are inversely modulated by AR activation, i.e., *SRD5A1* transcription is activated but *SRD5A2* is repressed, favoring a transcriptional isoform switch that correlates with SRD5A expression patterns in human tumors [11]. In metastatic prostate cancer (PCa) models, *in vitro* studies have shown that SRD5A1 is upregulated in the Met CRPC setting and is directly regulated by androgens [11–13]. In addition, patients carrying the more active GG genotype in *SRD5A2* rs523349 exhibited a higher risk of the progression and death, suggesting that high 5α-reductase activity due to *SRD5A2* rs523349 polymorphism may contribute to resistance to ADT in CRPC [14]. The mechanism underlying this transcriptional regulatory switch is still unknown.

In benign prostatic tissue, expression of SRD5A2 protein is variable and negatively correlated with methylation of the *SRD5A2* promoter. Our recent studies show that 30% of adult prostates without malignancy do not express the *SRD5A2* gene or protein, and that somatic suppression of *SRD5A2* during adulthood is dependent on epigenetic changes associated with methylation of the promoter region of the *SRD5A2* gene [15–17]. In this study, we tested *SRD5A2* promoter methylation of 42 prostatic specimens and 12 metastatic biopsies and found significant hypermethylation of *SRD5A2* promoter region for CRPC compared with benign prostatic specimens. Contrary to common belief that DNA methylation occurs during tumor initiation and progression [18], we demonstrated that hypermethylation of *SRD5A2* in the whole promoter region (cytosine-phosphate-guanine [CpG]# -72 to +65) was correlated with better OS and progression free survival (PFS) of CRPC patients, and hypermethylation of promoter subset, CpG sites (CpG # -39 to -2), was best correlated with OS and PFS of CRPC patients. Our study suggests that *SRD5A2* methylation in promoter regions, specifically at

CpG# -39 to -2, a condition that favors an estrogenic as opposed to an androgenic milieu in the prostate, is significantly associated with better survival for CRPC patients treated with ADT. Recognition of epigenetic modifications of *SRD5A2*, which affects the prostatic hormonal environment, may affect the choices and sequence of available therapies for management of CRPC.

## Materials and methods

### Patient specimens

With the approval of institutional review board at Massachusetts General Hospital (MGH) and Dana-Farber Cancer Institute, a total of 58 CRPC specimens and 36 benign prostatic specimens were used for methylation testing. Written consent was obtained from the study participants for the two cohorts. The Local CRPC cohort patients were diagnosed as CRPC between 2004 and 2013. Forty two formalin-fixed paraffin-embedded (FFPE) prostatic specimens were collected from patients who underwent transurethral resection of the prostate (TURP) at MGH to relieve patients' urinary outflow obstruction from locally advanced PCa. All patients had received primary or secondary ADT. Pathology samples were reviewed by an expert genitourinary pathologist (CLW). As controls, 12 frozen benign prostatic specimens were collected from patients who underwent TURP for symptomatic benign prostatic hyperplasia (BPH) at MGH. The metastatic cohort included 12 metastatic biopsies that were collected from CRPC patients who were treated with abiraterone plus dutasteride (Clinical Trial NCT01393730). For comparison with this cohort, 24 benign prostatic specimens were collected from patients who underwent TURP for symptomatic BPH at MGH.

### DNA methylation test: Targeted Next-Gen Bisulfite Sequencing (tNGBS) technique

Test procedure is described in Supplementary Information.

### Immunohistochemistry

IHC was completed as previously described [16, 17]. More detail is described in the Supplementary Information.

### TCGA prostate cancer data mining

The data of *SRD5A2* promoter methylation and expression of SRD5A2 in normal prostatic specimens and primary PCa specimens from The Cancer Genome Atlas (TCGA) database: https://tcga-data.nci.nih.gov/tcga/ was extracted. The promoter methylation and expression of SRD5A2 between normal and tumor tissues were compared via the UALCAN website: http://ualcan.path.uab.edu [19]. The association between *SRD5A2* promoter methylation and SRD5A2 expression was analyzed via the MEXPRESS website: https://mexpress.be [20].

### Statistical analysis

Continuous variables were reported as the means ± SD. The mean differences of average methylation ratios of *SRD5A2* promoter between CRPC and benign prostatic tissues were compared using the Student's *t*-test. A heatmap was generated to compare the average methylation ratios between the two groups and identify the specific promoter regions with the most significant difference. R package "pheatmap" (version 1.0.12, https://cran.r-project.org/web/packages/pheatmap; R software 3.6.0) was used to draw the heatmap of methylation sites in the two cohorts, respectively. Those methylation sites were automatically clustered according to the

"Euclidean" distance. With a cutoff value of the ratio of *SRD5A2* promoter methylation, CRPC patients were further sub-grouped as *SRD5A2*-hypermethylation group and *SRD5A2*-hypo-methylation group. Cutoff values were determined by X-tail software (version 3.6.1; Yale University, New Haven, USA). OS and PFS data of the CRPC patients were retrieved and calculated, then the ratio of *SRD5A2* promoter methylation was correlated to the prognosis of disease with survival curve and correlation analysis. Hazard ratios (HRs) of *SRD5A2* promoter methylation in predicting the OS and PFS of CRPC patients receiving ADT were calculated using the Cox proportional hazards regression model and tested using the log-rank test. Multi-variable analyses were performed, including GS, PSA, and ratio of *SRD5A2* methylation as covariates. A P value of $< 0.05$ was considered as statistical significance. All the statistical analyses were conducted by R (version 3.5.2; The R Foundation, Vienna, Austria) and GraphPad Prism (version 7.00; GraphPad Software, Inc., La Jolla, USA).

## Results

### Hypermethylation of the SRD5A2 promoter region for CRPC

Several independent studies have shown that expression of SRD5A1 is increased and SRD5A2 is decreased in the transition from hormone-naive PCa to CRPC [1, 13, 21, 22]. However, the mechanism and clinical significance of changes associated with SRD5A2 in CRPC is poorly understood. To explore whether epigenetic changes of *SRD5A2* affects the therapeutic efficacy of ADT, we tested and compared *SRD5A2* promoter methylation ratios between CRPC and benign prostatic tissues using the NGBS technique. We found that the methylation ratios of main *SRD5A2* promoter regions were higher for CRPC prostatic tissues compared with benign prostatic tissues in the local CRPC cohort (Fig 1A). To identify methylation in the *SRD5A2* specific promoter region, we performed unsupervised cluster analysis of the ratio of *SRD5A2* promoter methylation on all CpG methylation sites. The CpG methylation sites were clustered into three modules, CpG# -72 to -42; CpG#-40 to -35 / CpG# -31 to -19; CpG# -34 to -32 / CpG# -18 to 65 (Fig 1C).

Validating our finding in Met CRPC tissues demonstrated a similar finding as the Local CRPC cohort, i.e., higher methylation ratio of *SRD5A2* for CRPC patients compared with that in the benign control (Fig 1B). In addition, unsupervised cluster analysis generated two modules, CpG# -72 to -35; CpG# -34 to -32 / CpG# -30 to 65 (Fig 1D). After analysis of the Local and Met CRPC cohorts, we identified CpG# -39 to 65 as the important region that is differentially methylated between CRPC vs. benign prostatic samples.

We further verified our finding with Student's *t*-test based quantitative analyses. The average methylation ratios of the whole *SRD5A2* promoter region (CpG#-72 to 65) and specific promoter region CpG#-39 to 65 of CRPC tissues were significantly higher than that of benign prostatic tissues in the Local CRPC cohort (Fig 1E and 1I) and in the Met CRPC cohort (Fig 1F and 1J). However, there was no significant difference in the average methylation ratios of region CpG#-72 to -40 between CRPC and benign control groups (Fig 1G and 1H).

### Higher ratio of SRD5A2 promoter methylation was correlated with better prognosis of CRPC patients receiving ADT

We next addressed whether the *SRD5A2* promoter methylation is correlated to the clinical efficiency of ADT for CRPC patients. CRPC patients were sub-grouped as the *SRD5A2* hyper-methylation group and the *SRD5A2* hypomethylation group using the best cutoff value of methylation ratios. In the Local CRPC cohort, data of the whole promoter region of *SRD5A2*

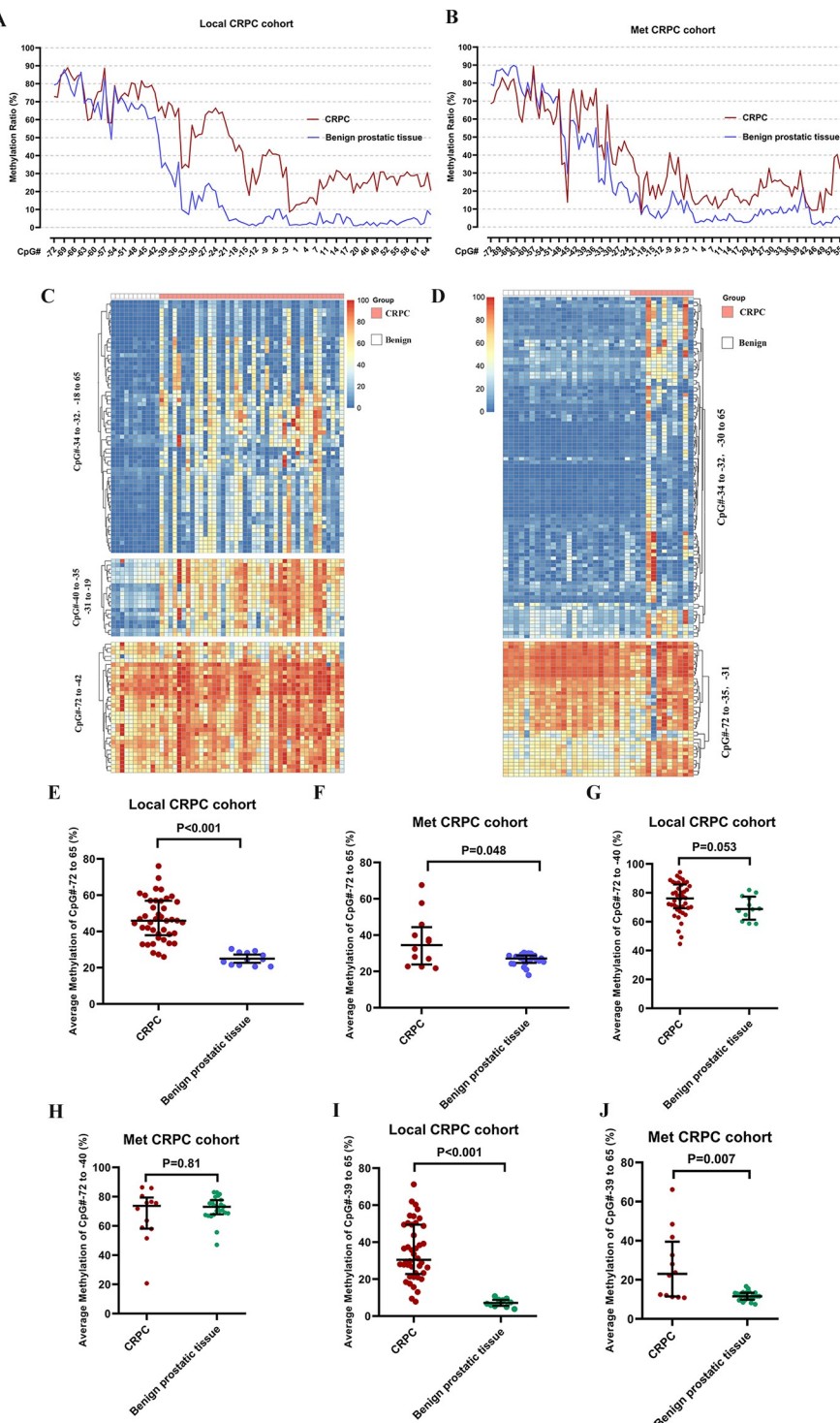

**Fig 1. Hypermethylation of the *SRD5A2* promoter region in CRPC.** (A) The ratio of *SRD5A2* promoter methylation on individual CpG sites in the promoter region in CRPC and benign prostatic tissues in the Local CRPC cohort. (B) The ratio of *SRD5A2* DNA promoter methylation on individual CpG sites in CRPC and benign prostatic tissues in the Met CRPC cohort. (C) Unsupervised cluster analysis of the ratio of *SRD5A2* promoter methylation on all CpG methylation sites in the Local CRPC cohort. The CpG methylation sites were clustered into 3 modules, CpG# -72 to -42; CpG# -40 to -35 and -31 to -19; CpG# -34 to -32 and -18 to 65. (D) Unsupervised cluster analysis of the ratio of *SRD5A2* promoter methylation on all the CpG methylation sites in the Met CRPC cohort. The methylation sites were clustered into 2 modules, CpG# -72 to -35, -31; CpG# -34 to -32 and -30 to 65. (E) The average methylation level of all

tested CpG methylation sites in CRPC and benign prostatic tissues in the Local CRPC cohort. (F) The average methylation level of all tested CpG methylation sites in CRPC and benign prostatic tissues in the Met CRPC cohort. (G) The average methylation level of module 1: CpG# -72 to -40 in CRPC and benign prostatic tissues in the Local CRPC cohort. (H) The average methylation level of module 1: CpG# -72 to -40 in CRPC and benign prostatic tissues in the Met CRPC cohort. (I) The average methylation level of module 2: CpG# -39 to 65 in CRPC and benign prostatic tissues in the Local CRPC cohort. (J) The average methylation level of module 2: CpG# -39 to 65 in CRPC and benign prostatic tissues in the Met CRPC cohort.

(CpG#-72 to 65) was generated as a Heatmap (Fig 2A), which revealed the different levels of methylation between the two groups.

Next, we retrieved OS and PFS data of the CRPC patients and correlated the ratio of *SRD5A2* promoter methylation to the prognosis of disease with a survival curve and correlation analysis. We found that a higher methylation ratio was significantly correlated with better OS of CRPC patients receiving ADT (S1 and S2 Tables, 12.0 ± 6.3 *vs* 8.0 ± 4.8 years, *P* = 0.013; Fig 2B and 2C) and better PFS (S1 and S2 Tables, 8.4 ± 5.6 *vs* 5.3 ± 4.2 years, *P* = 0.045; Fig 2D and 2E). Similarly, in the Met CRPC cohort, the Heatmap demonstrated the difference of methylation in whole promoter region of *SRD5A2* (CpG#-72 to 65) between the two groups (S2A Fig), and a higher methylation ratio was significantly associated with better OS (18.2 ± 5.2 *vs* 7.9 ± 4.3 years, *P* = 0.01, S2B and S2C Fig) and better PFS (10.0 ± 4.8 *vs* 5.0 ± 1.2 months, *P* = 0.002, S2D and S2E Fig). Together, our data suggest that *SRD5A2* whole promoter region methylation both in prostatic tissue and metastatic biopsies predicts a better outcome of ADT treatment for CRPC patients.

## Higher ratio of SRD5A2 methylation in specific promoter regions was correlated with better prognosis of CRPC patients receiving ADT

Next, we sought to investigate if the methylation in more specific *SRD5A2* promoter regions better predict the outcome of ADT treatment. We first focused our study on a specific promoter region CpG# -39 to 65 (Fig 1I and 1J). As expected, in the local CRPC cohort, a higher methylation ratio was significantly associated with better OS (*P* = 0.038, Fig 3A and 3B) and better PFS (*P* = 0.019, Fig 3C and 3D). Similarly, in the Met CRPC cohort, higher methylation was also significantly associated with better OS (*P* = 0.056, Fig 3E and 3F) and better PFS (*P* = 0.02, Fig 3G and 3H).

Furthermore, we expected to find the best promoter regions, which might be more predictive of the outcome of ADT treatment. Among the series of promoter regions, the region located at CpG# -39 to -2 had the most statistically significant value in the Local CRPC cohort (S1 Table). Higher methylation of this specific region (nucleotides -434 to -4 [CpG# -39 to CpG# -2]) was associated with better OS (11.3 ± 5.8 vs 6.4 ± 4.4 years, *P* = 0.001) and PFS (8.4 ± 5.4 vs 4.5 ± 3.9 years, *P* = 0.003) with a cutoff value of 37.9% (S1 Table and Fig 4). Similarly, in the Met CRPC cohort, higher methylation of this specific region was also significantly associated with OS (8.4 ± 4.3 vs 14.6 ± 8.2 years, *P* = 0.04) and PFS (5.0 ± 1.2 vs 10.0 ± 4.8 years, *P* = 0.002) with a cutoff value of 47% (S2 Table and S3 Fig). Our data suggest that hypermethylation in specific region, nucleotides -434 to -4 (CpG# -39 to CpG# -2) might be a good marker to predict the ADT sensitivity for CRPC patients.

In addition, multivariable analysis demonstrated that *SRD5A2* methylation both in the whole promoter region and specific region (CpG# -39 to -2) was associated with OS independent from GS and PSA (S3 Table, whole promoter region: *P* = 0.03; CpG# -39 to -2: *P* = 0.02).

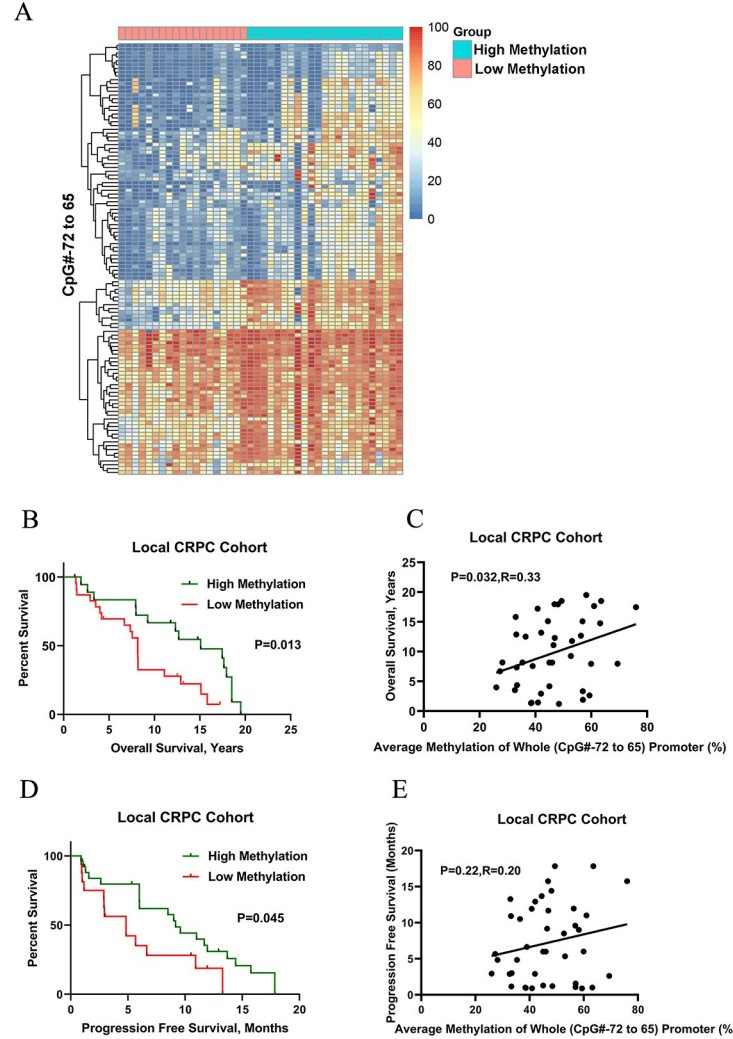

**Fig 2. Hypermethylation of *SRD5A2* in the whole promoter region (CpG # -72 to 65) was correlated with OS and PFS of CRPC patients in the Local CRPC cohort.** Patients were divided into a hypermethylation group and a hypomethylation group with a cutoff value of 46.5%, based on the average methylation level of all methylation sites. (A) Unsupervised cluster analysis of the ratio of *SRD5A2* promoter methylation on all CpG methylation sites. (B) The difference of OS between the hypermethylation group and hypomethylation group. (C) The correlation between methylation level and OS. (D & E) Patients were divided into a hypermethylation group and a hypomethylation group with a cutoff value of 42%, based on the average methylation level of all methylation sites. (D) The difference of PFS between the two groups. (E) The correlation between methylation level and PFS.

## SRD5A2 promoter methylation was negatively associated with SRD5A2 expression

Finally, we sought to investigate if *SRD5A2* promoter methylation is associated with SRD5A2 expression. IHC data demonstrated that the protein expression of SRD5A2 was negatively correlated with its methylation ratio both in the whole promoter region (CpG# -72 to CpG# 65) (R = 0.235, *P* = 0.0011) and specific region (CpG# -39 to CpG# -2) (R = 0.287, P = 0.0003) (Fig 5A–5C).

Then, we validated our finding using the publicly available TCGA dataset. The data of *SRD5A2* promoter methylation from 502 primary PCa tissues and 50 normal control prostatic

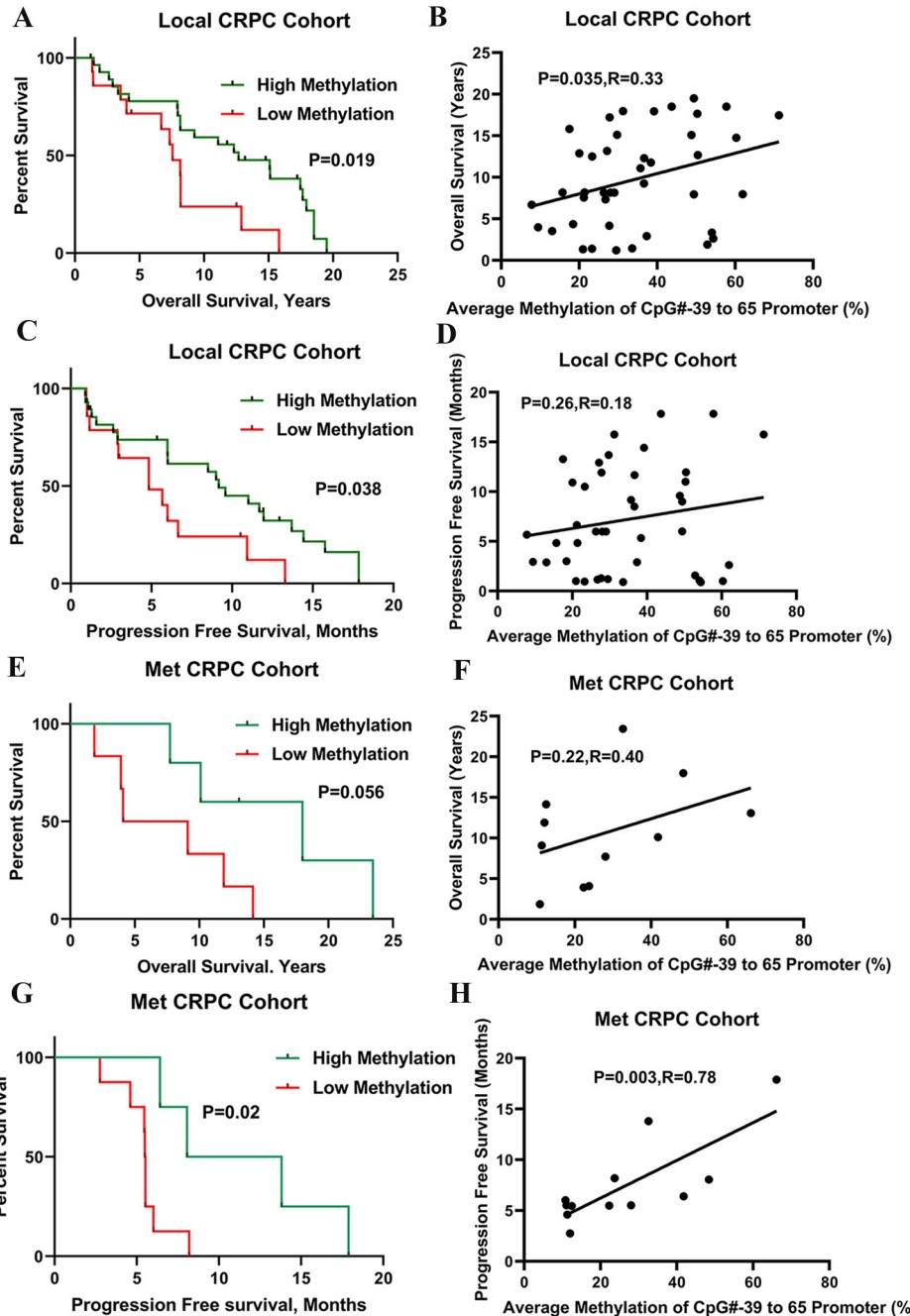

**Fig 3.** *SRD5A2* **hypermethylation of specific CpG sites (CpG # -39 to 65) in the promoter region was correlated with OS and PFS of CRPC patients.** Survival analysis was done based on clustering analysis of methylation site modules (CpG#-39 to 65 average methylation level). (A to D) In the Local CRPC cohort, patients were divided into a hypermethylation group and a hypomethylation group with a cutoff value of 23.3%, based on the average methylation level of specific CpG sites (CpG# -39 to 65). (A) The difference of OS between the two groups. (B) The correlation between *SRD5A2* promoter methylation level and OS. (C) The difference of PFS between the two groups. (D) The correlation between methylation level and PFS. (E to H) In the Met CRPC cohort, patients were divided into a hypermethylation group and a hypomethylation group with a cutoff value of 28.0%, based on the average methylation level of specific CpG sites (CpG# -39 to 65). (E) The difference of OS between the two groups. (F) The correlation between methylation level and OS. (G) The difference of PFS between the two groups. (H) The correlation between methylation level and PFS.

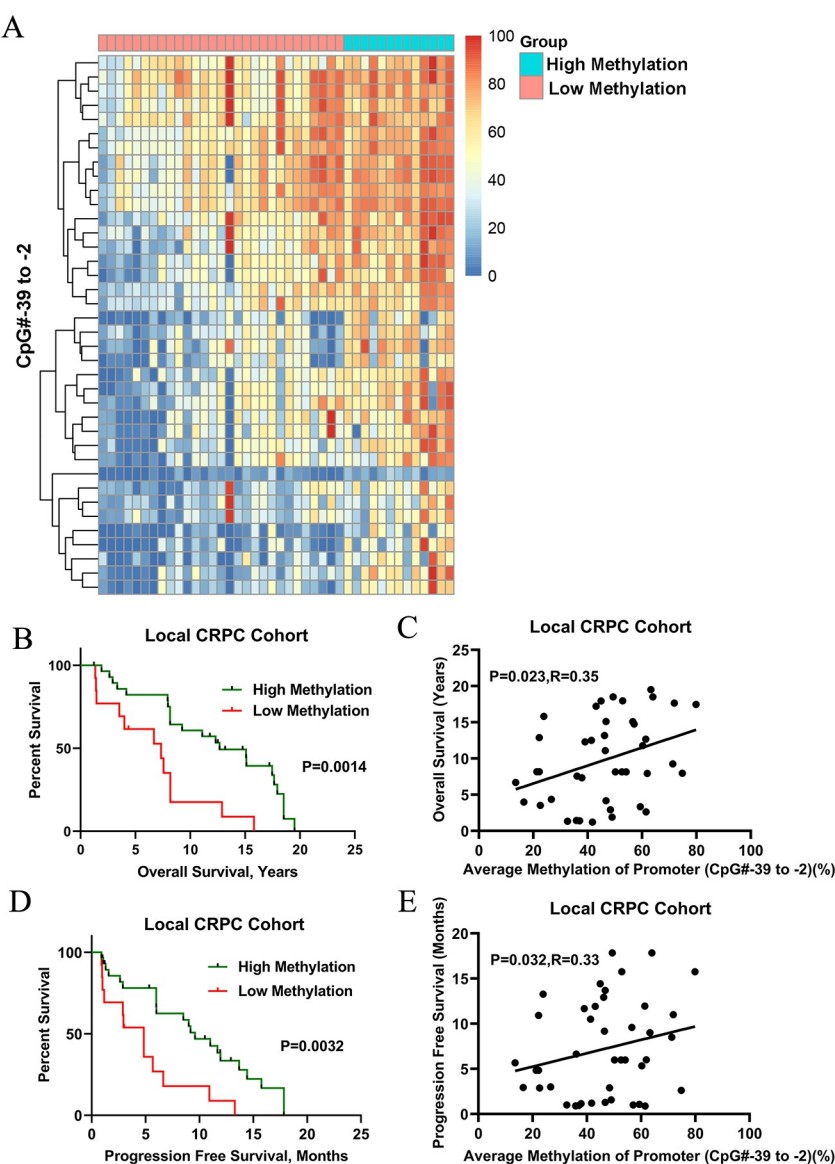

**Fig 4. *SRD5A2* hypermethylation of specific CpG sites (CpG # -39 to -2) in the promoter region was correlated with OS and PFS of CRPC patients in the Local CRPC cohort.** Patients were divided into a hypermethylation group and a hypomethylation group with a cutoff value of 37.9%, based on the average methylation level of specific CpG sites (CpG# -39 to -2). (A) Unsupervised cluster analysis of the ratio of *SRD5A2* promoter methylation on CpG methylation sites (CpG # -39 to -2). (B) The difference of OS between the hypermethylation group and hypomethylation group. (C) The correlation between methylation level and OS. (D) The difference of PFS between the two groups. (E) The correlation between *SRD5A2* methylation and PFS.

tissues were extracted from TCGA. We found that the promoter regions of *SRD5A2* of PCa specimen were significantly hypermethylated compared with normal prostatic tissues (P < 0.0001, Fig 5D). To compare the expression of SRD5A2, 497 primary PCa tissues and 52 normal prostatic tissues were analyzed, and we found that the expression of SRD5A2 in primary PCa tissues was significantly lower than the normal prostatic tissues (P < 0.0001, Fig 5E). Furthermore, we found that the methylation levels of most sites of the *SRD5A2* promoter were negatively correlated with expression of SRD5A2. Three of *SRD5A2* promoter sites had

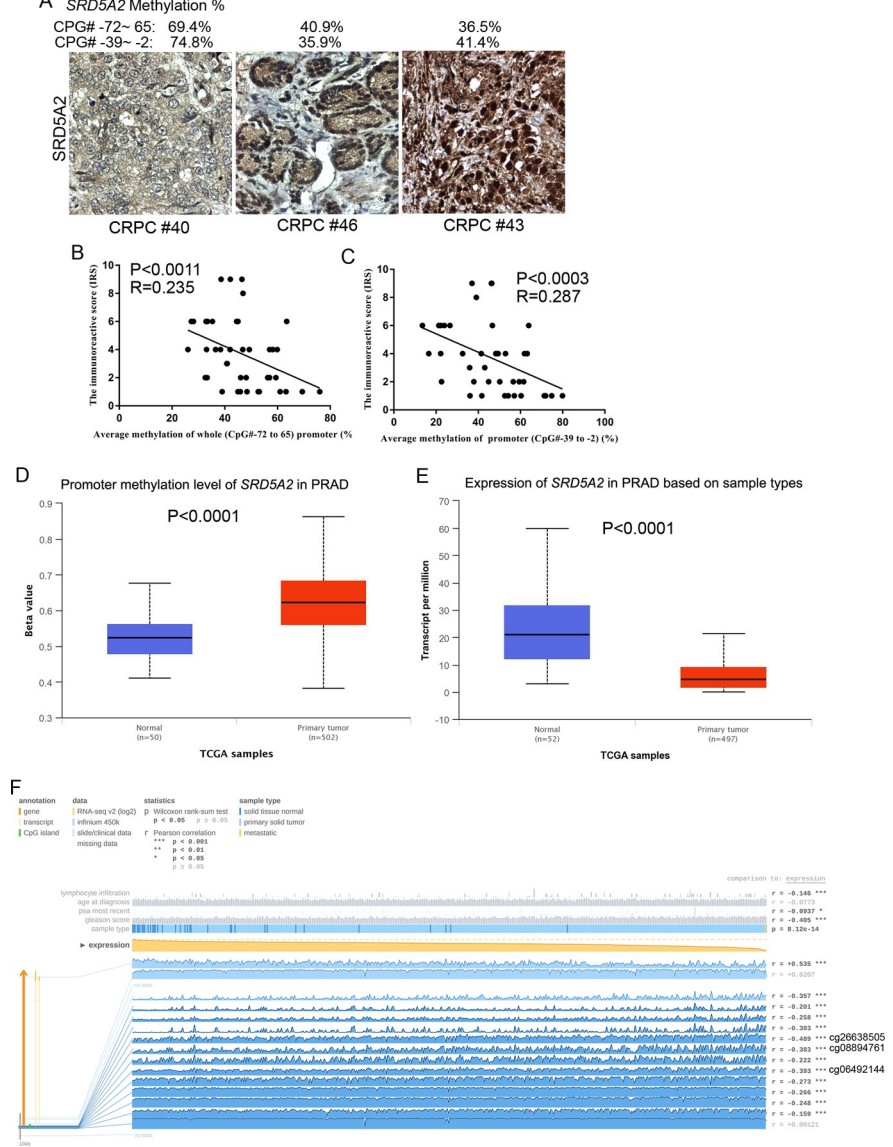

**Fig 5. Protein expression of SRD5A2 was negatively correlated with *SRD5A2* promoter methylation.** (A to C) IHC with anti-SRD5A2 antibody. (A) Representative pictures of IHC. (B) The immunoreactive score was correlated with *SRD5A2* promoter methylation on CpG# -72 to 65. (C) The immunoreactive score was correlated with *SRD5A2* promoter methylation on CpG# -39 to -2. (D to F) TCGA data analysis. (D) *SRD5A2* promoter methylation in primary PCa. (E) SRD5A2 expression in primary PCa. (F) The association between *SRD5A2* promoter methylation and SRD5A2 expression. Blue rows represented the methylation levels of *SRD5A2* promoter region, yellow rows represented SRD5A2 expression, and each column represented one sample.

the highest correlation with SRD5A2 expression, which were cg26638505 (CpG# -25, R = -0.489), cg06492144 (CpG# -35, R = -0.393) and cg08894761 (CpG# -25, R = -0.383), sites that overlap with CpG# -39 to CpG# -2 (Fig 5F). The data here further supports our finding that hypermethylation in specific region CpG# -39 to -2 might be a good marker to predict the ADT sensitivity for CRPC patients.

## Discussion

SRD5A2 is the predominant isoform of 5-α reductase expressed in the prostate, whereas SRD5A1 represents less than 10% of total 5α-reductase levels in normal prostate cells [23]. In PCa cells, the expression of SRD5A2 decreases, but the expression of SRD5A1 increases [21–24]. These two enzymes have been associated with PCa initiation and PCa progression. For instance, studies on polymorphisms in *SRD5A1* and *SRD5A2* have suggested its association with PCa risk and PCa recurrence [25–27]. However, *SRD5A2* rs9282858, the common nonsynonymous single nucleotide polymorphism was found not significantly associate with PCa risk [25]. Therefore, further studies are warranted to identify how the epigenetic change of these enzymes effects on PCa. To our knowledge, ours is the first study to analyze the association of epigenetic change of *SRD5A2* with response to ADT for CRPC patients. In contradistinction to widely held assumptions that DNA methylation is correlated with cancer progression, our findings show that hypermethylation in the *SRD5A2* promoter region, specifically CpG# -39 to -2, predicted better OS and PFS for patients with CRPC. The study suggests that quantitative *SRD5A2* methylation analysis in a pre-treatment biopsy could allow identification of patients most likely to benefit and facilitate tailoring of ADT therapy. For example, PCa patients with *SRD5A2* methylation may benefit less from intensified AR-directed therapy, or from early addition of docetaxel in castration-sensitive metastatic disease [28, 29]. Alternatively, patients with *SRD5A2* methylation may respond differently to androgen signaling inhibitors or targeted therapies toward the estrogen receptor pathway may prove beneficial in this subset of patients.

Tumor cells can be activated by epigenetic alterations, and further use epigenetic processes to escape from chemotherapy and host immune surveillance. Therefore, targeting the epigenome, including DNA methylation and histone modifications has been a growing emphasis of recent drug discovery [18]. In our study, data from two cohorts demonstrated that there is hypermethylation of *SRD5A2* promoter in CRPC compared with benign prostatic tissues (Fig 1). This result was further confirmed by analyzing TCGA data (Fig 5). Most interestingly, we found that *SRD5A2* methylation in promoter regions, specifically at CpG# -39 to CpG# -2, is significantly associated with better survival for CRPC patients treated with ADT. Being consistent with the study for high-risk gliomas which showed that *MGMT* promoter methylation predicts better survival outcomes [30], our finding not only provides evidence that alteration of the epigenome is an important step in cancer progression, but also opens new windows of opportunities in personalized medicine for CRPC patients.

There are several methods to test for DNA methylation or a CpG island in the promoter region of a single gene. The most widely used methods include pyrosequencing, methylation-specific polymerase chain reaction (PCR), and direct Sanger sequencing. In this study we used the tNGBS technique to test the methylation of the *SRD5A2* promoter region, which allowed us to analyze massive quantities of CpG sites for methylation [31, 32]. While we analyzed methylation of single CpG site in the *SRD5A2* promoter region, our future study will identify whether single CpG site(s) methylation within a CpG island shore of the *SRD5A2* gene contribute(s) to alteration in SRD5A2 expression and subsequent acquisition of treatment resistance.

Here, we present evidence that *SRD5A2* promoter methylation could be used as a prognostic marker for CRPC patients treated with ADT. An explanation could be that *SRD5A2* promoter hypermethylation causes the absence of SRD5A2 expression, and an androgenic to estrogenic switch in prostate tissue. This hypothesis is supported by our recent studies and others [17, 33–35]. In BPH tissues, we showed that the level of estradiol is dramatically elevated in prostatic samples with methylation of the *SRD5A2* promoter, a condition that favors an estrogenic, as opposed to an androgenic, milieu in the prostate. The phosphorylation of estrogen

receptor-α (ERα) in prostatic stroma is upregulated when SRD5A2 expression is absent [17]. During PCa initiation, not only androgens but also estrogens are required for malignant transformation [34, 35], suggesting that there is a role for AR/ERα cooperation in androgen targeted therapy resistance. Therefore, combination of treatments targeting both AR and ERα pathways may improve therapies for management of CRPC [34, 36]. Together, we postulate that *SRD5A2* methylation in promoter regions, a condition that favors an estrogenic, as opposed to an androgenic milieu in the prostate, is significantly associated with better survival for CRPC patients treated with ADT. Further studies are warranted to examine whether hormonal interventions play a role in CRPC disease progression and treatment efficiency.

## Supporting information

**S1 Fig. *SRD5A2* DNA methylation profiling.**
(PPTX)

**S2 Fig. Hypermethylation of *SRD5A2* in the whole promoter region (CpG # -72 to 65) was correlated with overall survival (OS) and progression free survival (PFS) of castration-resistant prostate cancer (CRPC) patients in the metastatic (Met) CRPC cohort.** Patients were divided into a hypermethylation group and a hypomethylation group with a cutoff value of 40.0%, based on the average methylation level of all methylation sites. (A) Unsupervised cluster analysis of the ratio of *SRD5A2* promoter methylation on all CpG methylation sites. (B) The difference of OS between the hypermethylation group and the hypomethylation group. (C) The correlation between methylation level and OS. (D & E) Patients were divided into a hypermethylation group and a hypomethylation group with a cutoff value of 32.5%, based on the average methylation level of all methylation sites. (D) The difference of PFS between hypermethylation group and hypomethylation group. (E) The correlation between *SRD5A2* promoter methylation level and PFS.
(TIF)

**S3 Fig. In the Met CRPC cohort, patients were divided into a hypermethylation group and a hypomethylation group with a cutoff value of 47%, based on the average methylation level of specific CpG sites (CpG# -39 to -2).** (A) Unsupervised cluster analysis of the ratio of *SRD5A2* promoter methylation. (B) The difference of OS between the two groups. (C) The correlation between methylation level and OS. (D & E) Patients were divided into a hypermethylation group and a hypomethylation group with a cutoff value of 37.8%, based on the average methylation level of specific CpG sites (CpG# -39 to -2). (D) The difference of PFS between the two groups. (E) The correlation between methylation level and PFS.
(TIF)

**S1 Material and methods. Supporting material and methods [16,17,37]**
(DOCX)

**S1 Protocol.**
(DOCX)

**S1 Table. Cutoff values of *SRD5A2* promoter methylation in Local CRPC cohort.**
(DOCX)

**S2 Table. Cutoff values of *SRD5A2* promoter methylation in Met CRPC cohort.**
(DOCX)

**S3 Table. Multivariable regression analyses.**
(DOCX)

## Author Contributions

**Conceptualization:** Zongwei Wang, Aria F. Olumi.

**Data curation:** Zongwei Wang, Tuo Deng, Xingbo Long, Shulin Wu.

**Formal analysis:** Zongwei Wang, Tuo Deng, Xingbo Long, Shulin Wu.

**Funding acquisition:** Aria F. Olumi.

**Investigation:** Tuo Deng, Xueming Lin, Hongbo Wang.

**Methodology:** Xueming Lin, Hongbo Wang, Rongbin Ge.

**Project administration:** Aria F. Olumi.

**Resources:** Zhenwei Zhang, Chin-Lee Wu, Mary-Ellen Taplin.

**Writing – original draft:** Zongwei Wang.

**Writing – review & editing:** Aria F. Olumi.

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
