## [Decision Letter · Decision Letter 0]

15 Jan 2020

PONE-D-19-33856

Methylation of SRD5A2 promoter predicts a better outcome for castration-resistant prostate cancer patients undergoing androgen deprivation therapy

PLOS ONE

Dear Dr. Olumi:

Thank you for submitting your manuscript to PLOS ONE. After careful consideration, we feel that it has merit and my decision is "**Minor Revision"** to fully meet  the PLOS ONE’s publication criteria (as it currently stands). Therefore, we invite you to submit a revised version of the manuscript that addresses the points raised during the review process.

**Reviewer #1:**

Understanding prostate cancer tumor progression mechanism and potential targets for treatment is very important. The 5a-Reductase types 1 and 2, encoded by SRD5A1 and SRD5A2, are key enzymes that catalyze the conversion of testosterone to dihydrotestosterone, androgen receptor (AR) agonist in prostate cells. 5a-Reductase type 2 is the predominant isoform expressed in the normal prostate. However, its expression decreases during prostate cancer (PCa) progression, whereas SRD5A1 increases, and the mechanism underlying this transcriptional regulatory switch is still unknown. In this research manuscript, Wang, et al  reports that methylation of *SRD5A2* promoter occurs frequently in CRPC patients treated with ADT, and provided  important patient data (from local cohorts and TCGA). Furthermore, authors demonstrated the negative correlation between promoter methylation of SRD5A2 and SRD5A2 expression.

**Comments:**

1. The major strength of the study is the patient data from two cohorts to validate their hypothesis and comparison of findings with the TCGA data. Paper is well written and all the experiments with perfect controls in place.

2. Authors shall provide explanation about the basis on which CpG methylation sites were clustered into 3 modules, (CpG# -72 to -42; CpG# -40 to -35 and -31 to -19; CpG# -34 to -32 and -18 to 65 in CRPC patients) whereas methylation sites were clustered into 2 modules (CpG# -72 to -35, -31; CpG# -34 to -32 and -30 to 65 in metastatic CRPC cohorts).

3. Have the authors found any relation between the expression of SRD5A2 and survival in local CRPC patient cohorts, although they have shown through the TCGA data analyzed by MEXPRESS?

4. In Figure 3, 4 and 5 only 3 H has a value of “r”=0.78 (positive correlation), in other panels, “r” shows weak correlation, would the authors explain why.

5. In Figure 5 B and C, “r” and “P” values are not shown. Authors shall add the “r” and “P” in figure 5 B and C.

6. Have authors found any correlation between Androgen Receptor expression and SRD5A2 expression in local patient cohorts?

7. Authors have used local patent cohorts, please provide a link where data can be accessible if possible as per the PLosOne data sharing policy.

**Reviewer #2: **

The present study is a provacative analysis on the methylation status of the SDR5A2 promoter, which encodes an enzyme that metabolizes testosterone to the more potent androgen DHT, in castrate-resistant prostate cancer (CRPC)( specimens. Although the numbers of specimens analyzed in the present data are relatively small, the findings do show that there is a significant trend for promoter hypermethylation in CRPC specimens compared to benign prostate tissue. Most significantly, the higher ratio of promoter hypermethylation was positively associated with a significant increase in time to progression and patient survival when the patient data was interrogated. Immunohistochemistry analysis of SDR5A2 protein levels inversely correlated with the gene methylation levels, although this data has marked overlap and requires further follow-up. The data were further studied in the TCGA database which supported the primary data in this manuscript. Although the number of metastatic samples was very low in this study (n=12), the data indicate that the SDR5A2 methylation status is likewise able to predict patient survival.  The data are well discussed and fit well into emerging findings regarding the altered hormonal milieu in CRPC that drives androgen resistance.

**Comments:**

1. The paper would be improved by cell-based interrogations that replicate the specimen-based observations and support the conclusions, i.e. modifications of SDR5A2 methylation status change expression of this gene and alter its protein levels as well as steroid metabolism.

2. At present, the studies are merely correlative and direct studies are warranted. 

We would appreciate receiving your revised manuscript by Feb 28 2020 11:59PM. To enhance the reproducibility of your results, we recommend that if applicable you deposit your laboratory protocols in protocols.io, where a protocol can be assigned its own identifier (DOI) such that it can be cited independently in the future. For instructions see: http://journals.plos.org/plosone/s/submission-guidelines#loc-laboratory-protocols

We look forward to receiving your revised manuscript.

Kind regards,

**Mohammad Saleem,  **

University of Minnesota

2. We noticed minor instances of text overlap with the following previous publication(s), which need to be addressed:

(1) https://www.mdpi.com/2073-4425/9/9/429/htm

(2) https://www.sciencedirect.com/science/article/abs/pii/S0046817719300747?via%3Dihub

(3) https://pubs.acs.org/doi/10.1021/acs.molpharmaceut.7b00070

(4) https://onlinelibrary.wiley.com/doi/full/10.1002/cncr.30071

The text that needs to be addressed involves the Discussion section.

In your revision please ensure you cite all your sources (including your own works), and quote or rephrase any duplicated text outside the methods section. Further consideration is dependent on these concerns being addressed.

3. Please provide additional details regarding participant consent.

In the ethics statement in the Methods and online submission information, please ensure that you have specified (i) whether consent was informed and (ii) what type you obtained (for instance, written or verbal, and if verbal, how it was documented and witnessed).

5. Please upload a copy of Figure 6, to which you refer in your text on page 16. If the figure is no longer to be included as part of the submission please remove all reference to it within the text.

6. Please include captions for your Supporting Information files at the end of your manuscript, and update any in-text citations to match accordingly. Please see our Supporting Information guidelines for more information: http://journals.plos.org/plosone/s/supporting-information

---

## [Author Response · Author response to Decision Letter 0]

24 Jan 2020

Reviewer #1:

Understanding prostate cancer tumor progression mechanism and potential targets for treatment is

very important. The 5a-Reductase types 1 and 2, encoded by SRD5A1 and SRD5A2, are key enzymes

that catalyze the conversion of testosterone to dihydrotestosterone, androgen receptor (AR) agonist in

prostate cells. 5a-Reductase type 2 is the predominant isoform expressed in the normal prostate.

However, its expression decreases during prostate cancer (PCa) progression, whereas SRD5A1

increases, and the mechanism underlying this transcriptional regulatory switch is still unknown. In this

research manuscript, Wang, et al reports that methylation of SRD5A2 promoter occurs frequently in

CRPC patients treated with ADT, and provided important patient data (from local cohorts and TCGA).

Furthermore, authors demonstrated the negative correlation between promoter methylation of SRD5A2

and SRD5A2 expression.

Comments:

1. The major strength of the study is the patient data from two cohorts to validate their hypothesis and

comparison of findings with the TCGA data. Paper is well written and all the experiments with perfect

controls in place.

Response: Thanks for reviewer’s positive comments.

2. Authors shall provide explanation about the basis on which CpG methylation sites were clustered

into 3 modules, (CpG# -72 to -42; CpG# -40 to -35 and -31 to -19; CpG# -34 to -32 and -18 to 65 in

CRPC patients) whereas methylation sites were clustered into 2 modules (CpG# -72 to -35, -31; CpG#

-34 to -32 and -30 to 65 in metastatic CRPC cohorts).

Response: Thanks for this suggestion. In the section of Statistical Analysis, we explained: R

package “pheatmap” which was used to draw the heatmap of methylation sites in the two

cohorts, respectively. Those methylation sites were automatically clustered according to the

“Euclidean” distance.

3. Have the authors found any relation between the expression of SRD5A2 and survival in local CRPC

patient cohorts, although they have shown through the TCGA data analyzed by MEXPRESS?

Response: Our two cohorts include patients with metastatic CRPC. As a result, we have

focused our study on the methylation of SRD5A2 promoter and survival in patients with

metastatic CRPC. To address the reviewer’s concern, as mentioned by the reviewer, we utilized

the TCGA data to assess SRD5A2’s promoter methylation. In the near future studies, we plan to

evaluate if the expression of SRD5A2 is associated with survival of localized CRPC patients.

4. In Figure 3, 4 and 5 only 3H has a value of “r”=0.78 (positive correlation), in other panels, “r” shows

weak correlation, would the authors explain why.

Response: Thanks for this careful review. In Figure 3, 4 and 5, the majority of analyses showed

the significant correlations (P<0.05), which are consistent with the Kaplan Meier survival curve.

The low R value might be caused by the small number of subjects.

5. In Figure 5 B and C, “r” and “P” values are not shown. Authors shall add the “r” and “P” in figure 5 B

and C.

Response: Thanks for this suggestion. The “r” and “P” values in figure 5 B and C have been

added.

6. Have authors found any correlation between Androgen Receptor expression and SRD5A2

expression in local patient cohorts?

Response: Thanks for this question. At this moment, we did not evaluate the correlation

between AR and SRD5A2 expression.

7. Authors have used local patient cohorts, please provide a link where data can be accessible if

possible as per the PLosOne data sharing policy.

Response: Thanks for this suggestion. We have made the raw informatics data accessible

online. Link: https://doi.org/10.7910/DVN/MPAM2H

Reviewer #2:

The present study is a provacative analysis on the methylation status of the SDR5A2 promoter, which

encodes an enzyme that metabolizes testosterone to the more potent androgen DHT, in castrateresistant

prostate cancer (CRPC)( specimens. Although the numbers of specimens analyzed in the

present data are relatively small, the findings do show that there is a significant trend for promoter

hypermethylation in CRPC specimens compared to benign prostate tissue. Most significantly, the

higher ratio of promoter hypermethylation was positively associated with a significant increase in time to

progression and patient survival when the patient data was interrogated. Immunohistochemistry

analysis of SDR5A2 protein levels inversely correlated with the gene methylation levels, although this

data has marked overlap and requires further follow-up. The data were further studied in the TCGA

database which supported the primary data in this manuscript. Although the number of metastatic

samples was very low in this study (n=12), the data indicate that the SDR5A2 methylation status is

likewise able to predict patient survival. The data are well discussed and fit well into emerging findings

regarding the altered hormonal milieu in CRPC that drives androgen resistance.

Comments:

1. The paper would be improved by cell-based interrogations that replicate the specimen-based

observations and support the conclusions, i.e. modifications of SDR5A2 methylation status change

expression of this gene and alter its protein levels as well as steroid metabolism.

Response: Thanks for the reviewer’s great suggestion on the mechanism study.

Utilizing prostate cancer LnCap, DU145 and other cell lines, we are working to

understand whether and how the modification of SRD5A2 methylation status alters

steroid metabolism. This suggestion will be part of future publications which is some

time away.

2. At present, the studies are merely correlative and direct studies are warranted.

Response: Again, we thank the reviewer’s for his/her insight. Currently, we are using cultured

cells to identify whether there is a cause and effect relationship between SRD5A2 methylation

and better survival for CRPC patients using in-vitro and in-vivo model systems.

1. Please ensure that your manuscript meets PLOS ONE's style requirements, including those for file

naming. The PLOS ONE style templates can be found at

http://www.journals.plos.org/plosone/s/file?id=wjVg/PLOSOne_formatting_sample_main_body.pdf and

http://www.journals.plos.org/plosone/s/file?id=ba62/PLOSOne_formatting_sample_title_authors_affiliati

ons.pdf

Response: We have carefully reviewed and revised the manuscript to meet PLOS ONE's style

requirements.

2. We noticed minor instances of text overlap with the following previous publication(s), which need to

be addressed:

(1) https://www.mdpi.com/2073-4425/9/9/429/htm

Response: It has been quoted and rephrased. Please see page 21.

(2) https://www.sciencedirect.com/science/article/abs/pii/S0046817719300747?via%3Dihub

Response: It has been addressed. Please see page 22.

(3) https://pubs.acs.org/doi/10.1021/acs.molpharmaceut.7b00070

Response: This literature is not related to our study. Not sure where the text overlap is.

(4) https://onlinelibrary.wiley.com/doi/full/10.1002/cncr.30071

Response: It has been quoted and rephrased. Please see page 19.

The text that needs to be addressed involves the Discussion section.

In your revision please ensure you cite all your sources (including your own works), and quote or

rephrase any duplicated text outside the methods section. Further consideration is dependent on these

concerns being addressed.

Response: Thanks for this careful review. We have cited all the sources, quoted and rephrased

all the duplicated text.

3. Please provide additional details regarding participant consent.

In the ethics statement in the Methods and online submission information, please ensure that you have

specified (i) whether consent was informed and (ii) what type you obtained (for instance, written or

verbal, and if verbal, how it was documented and witnessed).

Response: We have specified in the Section of Patient specimen (page 8): Written consent was

obtained from the study participants for the two cohorts.

4. PLOS requires an ORCID iD for the corresponding author in Editorial Manager on papers submitted

after December 6th, 2016. Please ensure that you have an ORCID iD and that it is validated in Editorial

Manager. To do this, go to ‘Update my Information’ (in the upper left-hand corner of the main menu),

and click on the Fetch/Validate link next to the ORCID field. This will take you to the ORCID site and

allow you to create a new iD or authenticate a pre-existing iD in Editorial Manager. Please see the

following video for instructions on linking an ORCID iD to your Editorial Manager account:

https://www.youtube.com/watch?v=_xcclfuvtxQ

Response: We have added and validated the ORCID for the corresponding author.

5. Please upload a copy of Figure 6, to which you refer in your text on page 16. If the figure is no

longer to be included as part of the submission please remove all reference to it within the text.

Response: Thanks for this careful review. This is the typo, which should refer to figure 5 on

current page 20.

6. Please include captions for your Supporting Information files at the end of your manuscript, and

update any in-text citations to match accordingly. Please see our Supporting Information guidelines for

more information: http://journals.plos.org/plosone/s/supporting-information

Response: Captions for the Supporting Information files have been included at the end of

manuscript, and in-text citations have been updated accordingly.

---

## [Decision Letter · Decision Letter 1]

14 Feb 2020

Methylation of SRD5A2 promoter predicts a better outcome for castration-resistant prostate cancer patients undergoing androgen deprivation therapy

PONE-D-19-33856R1

Dear Dr. Olumi,

We are pleased to inform you that your manuscript has been judged scientifically suitable for publication and will be formally accepted for publication once it complies with all outstanding technical requirements. Within one week, you will receive an e-mail containing information on the amendments required prior to publication. When all required modifications have been addressed, you will receive a formal acceptance letter and your manuscript will proceed to our production department and be scheduled for publication.

With Kind Regards,

Mohammad Saleem, Academic Editor-PLOS ONE

University of Minnesota-Minneapolis

---

## [Editor Report · Acceptance letter]

19 Feb 2020

PONE-D-19-33856R1 

Methylation of *SRD5A2* promoter predicts a better outcome for castration-resistant prostate cancer patients undergoing androgen deprivation therapy 

Dear Dr. Olumi:

I am pleased to inform you that your manuscript has been deemed suitable for publication in PLOS ONE. Congratulations! Your manuscript is now with our production department. 

With kind regards,

on behalf of

Dr. MOHAMMAD Saleem 

Academic Editor

PLOS ONE